# ROBUST TEMPORAL ENSEMBLING

## ABSTRACT

Successful training of deep neural networks with noisy labels is an essential capability as most real-world datasets contain some amount of mislabeled data. Left unmitigated, label noise can sharply degrade typical supervised learning approaches. In this paper, we present *robust temporal ensembling* (RTE), a simple supervised learning approach which combines robust task loss, temporal pseudo-labeling, and a ensemble consistency regularization term to achieve noise-robust learning. We demonstrate that RTE achieves state-of-the-art performance across the CIFAR-10, CIFAR-100, and ImageNet datasets, while forgoing the recent trend of label filtering/fixing. In particular, RTE achieves 93.64% accuracy on CIFAR-10 and 66.43% accuracy on CIFAR-100 under 80% label corruption, and achieves 74.79% accuracy on ImageNet under 40% corruption. These are substantial gains over previous state-of-the-art accuracies of 86.6%, 60.2%, and 71.31%, respectively, achieved using three distinct methods. Finally, we show that RTE retains competitive corruption robustness to unforeseen *input* noise using CIFAR-10-C, obtaining a mean corruption error (mCE) of 13.50% even in the presence of an 80% noise ratio, versus 26.9% mCE with standard methods on clean data.

## 1 INTRODUCTION

Deep neural networks have enjoyed considerable success across a variety of domains, and in particular computer vision, where the common theme is that more labeled training data yields improved model performance (Hestness et al., 2017; Mahajan et al., 2018; Xie et al., 2019b; Kolesnikov et al., 2019). However, performance depends on the quality of the training data, which is expensive to collect and inevitably imperfect. For example, ImageNet (Deng et al., 2009) is one of the most widely-used datasets in the field of deep learning and despite over 2 years of labor from more than 49,000 human annotators across 167 countries, it still contains erroneous and ambiguous labels (Fei-Fei & Deng, 2017; Karpathy, 2014). It is therefore essential that learning algorithms in production workflows leverage noise robust methods.

Noise robust learning has a long history and takes many forms (Natarajan et al., 2013; Frenay & Verleysen, 2014; Song et al., 2020). Common strategies include loss correction and reweighting (Patrini et al., 2016; Zhang & Sabuncu, 2018; Menon et al., 2020), label refurbishment (Reed et al., 2014; Song et al., 2019), abstention (Thulasidasan et al., 2019), and relying on carefully constructed *trusted* subsets of human-verified labeled data (Li et al., 2017; Hendrycks et al., 2018; Zhang et al., 2020). Additionally, recent methods such as SELF (Nguyen et al., 2020) and DivideMix (Li et al., 2020) convert the problem of learning with noise into a semi-supervised learning approach by splitting the corrupted training set into clean labeled data and noisy unlabeled data at which point semi-supervised learning methods such as Mean Teacher (Tarvainen & Valpola, 2017) and MixMatch (Berthelot et al., 2019) can be applied directly. In essence, these methods effectively discard a majority of the label information so as to side-step having to learning with noise at all. The problem here is that noisy label filtering tactics are imperfect resulting in corrupted data in the small labeled partition and valuable clean samples lost to the large pool of unlabeled data. Moreover, caution is needed when applying semi-supervised methods where the labeled data is not sampled i.i.d. from the pool of unlabeled data (Oliver et al.). Indeed, filtering tactics can be biased and irregular, driven by specification error and the underlying noise process of the label corruption. Recognizing the success of semi-supervised approaches, we ask: can we leverage the underlying mechanisms of semi-supervised learning such as entropy regularization for learning with noise without discarding our most valuable asset, the labels?

## 2 ROBUST TEMPORAL ENSEMBLING

### 2.1 PRELIMINARIES

Adopting the notation of Zhang & Sabuncu (2018), we consider the problem of classification where $\mathcal{X} \subset \mathbb{R}^d$ is the feature space and $\mathcal{Y} = \{1, \ldots, c\}$ is the label space where the classifier function is a deep neural network with a softmax output layer that maps input features to distributions over labels $f : \mathcal{X} \to \mathbb{R}^c$. The dataset of training examples containing *in-sample* noise is defined as $D = \{(x_i, \tilde{y}_i)\}_{i=1}^n$ where $(x_i, \tilde{y}_i) \in (\mathcal{X} \times \mathcal{Y})$ and $\tilde{y}_i$ is the noisy version of the true label $y_i$ such that $p(\tilde{y}_i = k | y_i = j, x_i) \equiv \eta_{ijk}$. We do not consider *open-set* noise (Wang et al., 2018), in which there is a particular type of noise that occurs on inputs, $\tilde{x}$, rather than labels. Following most prior work, we make the simplifying assumption that the noise is conditionally independent of the input, $x_i$, given the true labels. In this setting, we can write $\eta_{ijk} = p(\tilde{y}_i = k | y_i = j) \equiv \eta_{jk}$ which is, in general, considered to be *class dependent* noise[1,2].

To aid in a simple and precise corruption procedure, we now depart from traditional notation and further decompose $\eta_{jk}$ as $p_j \cdot c_{jk}$, where $p_j \in [0, 1]$ is the probability of corruption of the $j$-th class and $c_{jk} \in [0, 1]$ is the relative probability that corrupted samples of class $j$ are labeled as class $k$, with $c_{i \neq j} \geq 0$, $c_{jj} = 0$ and $\sum_k c_{jk} = 1$. A noisy dataset with $m$ classes can then be described as transition probabilities specified by

$$F = \mathrm{diag}(P) \cdot C + \mathrm{diag}(1 - P) \cdot \mathcal{I} \tag{1}$$

where $C \in \mathbb{R}^{m \times m}$ defines the system confusion or noise structure, $P \in \mathbb{R}^m$ defines the noise intensity or ratio for each class, and $\mathcal{I}$ is the identity matrix. When $c_{jk} = c_{kj}$ the noise is said to be *symmetric* and is considered *asymmetric* otherwise. If ratio of noise is the same for all classes then $p_j = p$ and the dataset is said to exhibit *uniform* noise. For the case of uniform noise, equation (1) interestingly takes the familiar form of the Google matrix equation as

$$F_p = p \cdot C + (1 - p) \cdot \mathcal{I} \tag{2}$$

Note that, by this definition, $\eta_{jj} = p \cdot c_{jj} = 0$ which prohibits $\tilde{y}_i = y_i$. This ensures a true effective noise ratio of $p$. For example, suppose there are $m = 10$ classes and we wish to corrupt labels with 80% probability. Then if corrupted labels are sampled from $\mathcal{Y}$ rather than $\mathcal{Y} \setminus \{y\}$, $\frac{1}{10} \cdot 0.8 = 8\%$ of the corrupted samples will not actually be corrupted, leading to a *true* corruption rate of 72%. Therefore, despite prescribing $p = 0.8$, the true effective noise ratio would be 0.72, which in turn yields a $\frac{0.08}{1-0.8} = 40\%$ increase in clean labels, and this is indeed the case in many studies (Zhang & Sabuncu, 2018; Nguyen et al., 2020; Li et al., 2020; Zhang et al., 2020).

### 2.2 METHODS

At a very high level, RTE is the combination of noise-robust task loss, augmentation, and pseudo-labeling for consistency regularization. We unify generalized cross entropy (Zhang & Sabuncu, 2018), AugMix stochastic augmentation strategy (Hendrycks et al., 2020), an exponential moving average of model weights for generating pseudo-labels (Tarvainen & Valpola, 2017), and an augmentation anchoring-like approach (Berthelot et al., 2020) to form a robust approach for learning with noisy labels.

#### 2.2.1 NOISE-ROBUST TASK LOSS

Generalized cross entropy (GCE) (Zhang & Sabuncu, 2018) is a theoretically grounded noise-robust loss function that can be seen as a generalization of mean absolute error (MAE) and categorical cross entropy (CCE). The main idea is that CCE learns quickly, but more emphasis is put on difficult samples which is prone to overfit noisy labels, while MAE treats all samples equally, providing noise-robustness but learning slowly. To exploit the benefits of both MAE and CCE, a negative Box-Cox transformation (Box & Cox, 1964) is used as the loss function

$$\mathcal{L}_q(f(x_i), y_i = j) = \frac{(1 - f_j(x_i)^q)}{q} \tag{3}$$

---

[1]See Lee et al. (2019) for treatment of conditionally dependent *semantic* noise such that $\eta_{ijk} \neq \eta_{jk}$.

[2]Note that Patrini et al. (2016) define the noise transition matrix $T$ such that $T_{jk} \equiv \eta_{jk}$.

where $q \in (0, 1]$, and $f_j$ denotes the $j$-th element of $f$. Note that GCE becomes CCE for $\lim_{q \to 0} \mathcal{L}_q$ and becomes MAE/unhinged loss when $q = 1$.

### 2.2.2 ENSEMBLE CONSISTENCY REGULARIZATION

Consistency regularization works under the assumption that a model should output similar predictions given augmented versions of the same input. This regularization strategy is a common component of semi-supervised learning algorithms with the general form of $\|p_\theta(y|x_{\text{aug1}}) - p_\theta(y|x_{\text{aug2}})\|$ where $p_\theta(y|x)$ is the predicted class distribution produced by the model having parameters $\theta$ for input $x$ (Zheng et al., 2016; Sajjadi et al., 2016). We build upon numerous variations from semi-supervised learning (Laine & Alia, 2017; Tarvainen & Valpola, 2017; Berthelot et al., 2019; 2020) and leverage an ensemble consistency regularization (ECR) strategy as

$$\text{ECR} = \frac{1}{|\mathcal{Y}|N^*} \sum_{i=1}^{N^*} \|p_{\theta'}(y|x) - p_\theta(y|\mathcal{A}(x))\| \tag{4}$$

where $x$ is the training example, $\mathcal{A}$ is stochastic augmentation function reevaluated for each term in the summation, $\theta'_t = \alpha\theta'_{t-1} + (1 - \alpha)\theta_t$ is a temporal moving average of model weights used to generate pseudo-label targets, and inputs are pre-processed with standard random horizontal flip and crop. In practice, this consists of initializing a copy of the initial model and maintaining an exponential moving average as training progresses. Some methods directly average multiple label predictions together at each optimization step to form a single pseudo-label target (Berthelot et al., 2019; Li et al., 2020) but we find pseudo-label target distributions generated by $\theta'$ to be better suited for the learning with noise problem due to the intrinsic ensemble nature of the weight averaging process over many optimization steps (Tarvainen & Valpola, 2017). In semi-supervised learning techniques, it is common to leverage a large batch-size of unlabeled data for consistency regularization. However, we found that modulating $N^*$, rather than the batch size of the consistency term, yields a monotonic increase in model performance consistent with related works (Berthelot et al., 2020). Moreover, in semi-supervised learning, different batches are used for between supervised and unsupervised loss terms but we find (see section 4.3) that for the case of learning with noise, batches synchronized with GCE task loss term yields superior performance.

### 2.2.3 AUGMENTATION

AugMix (Hendrycks et al., 2020) is a data augmentation technique which utilizes stochasticity, diverse augmentations, a Jensen-Shannon divergence consistency loss, and a formulation to mix multiple augmented inputs. Other augmentation strategies such as RandAugment (Cubuk et al., 2020), augmentations are applied sequentially with fixed intensity which can degrade input quickly. In AugMix, to mitigate input degradation but retain augmentation diversity, several stochastically sampled augmentation chains are layered together in a convex combination to generate highly diverse transformations. These mixing coefficients are randomly sampled from a Dirichlet distribution with shared concentration parameters, and the resulting augmented version of the input is combined with the original input through a second random convex combination sampled from a beta distribution, again with shared parameters.

### 2.2.4 JENSEN-SHANNON DIVERGENCE

The Jensen-Shannon consistency loss is used to enforce a flat response of the classifier by incentivizing the model to be stable, consistent, and insensitive across a diverse range of inputs (Zheng et al., 2016). The Jensen-Shannon divergence (JSD) is minimized across distributions $p_{\text{orig}}$, $p_{\text{aug1}}$, and $p_{\text{aug2}}$ of the original sample $x_{\text{orig}}$ and its augmented variants $x_{\text{aug1}}$ and $x_{\text{aug2}}$ which can be understood to measure the average information that the sample reveals about the identity of its originating distribution (Hendrycks et al., 2020). This JSD term is computed with $M = (p_{\text{orig}} + p_{\text{aug1}} + p_{\text{aug2}})/3$ and is then

$$\text{JSD} = \frac{1}{3}\Big(\text{KL}(p_{\text{orig}} \| M) + \text{KL}(p_{\text{aug1}} \| M) + \text{KL}(p_{\text{aug2}} \| M)\Big) \tag{5}$$

where $\text{KL}(p \| q)$ is Kullback–Leibler divergence from $q$ to $p$. In this way, the JSD term improves the stability of training in the presence of noisy labels and heavy data augmentation with a modest contribution to final classifier test accuracy as shown in Table 5.

## 2.3 Putting It All Together

We unify the various components defined in sections 2.2 together under a single parsimonious loss function at training defined as

$$L_{\text{RTE}} = \mathcal{L}_q + \lambda_{\text{JSD}} \cdot \text{JSD} + \lambda_{\text{ECR}} \cdot \text{ECR} \tag{6}$$

which is essentially composed of robust task loss and consistency regularization. Here the JSD term is synchronized with ECR by computing the clean distribution using $p_{\theta'}$. Final performance is reported using $\theta'$.

To understand the synergy of GCE and ECR it is helpful to first point out that because GCE leverages a Box-Cox power transform to stabilize loss variance, it can be shown in this case to be a form of maximum likelihood estimation (Ferrari & Yang, 2010). The ECR term itself is based on pseudo-labeling and pseudo-labeling can be shown to be form of entropy regularization (Grandvalet & Bengio, 2004) which in the framework of maximum a posterior (MAP) estimation encourages low-density separation between classes by minimizing the conditional entropy of the class probabilities of the noisy data (Lee, 2013). That is, by minimizing entropy, the overlap of class probability distribution can be reduced. The implicit assumption here is that classes are, in fact, well separated (Chapelle & Zien, 2005). Moreover, MAP estimation itself acts as a regularization of MLE by incorporating a priori knowledge of related training examples in order to solve the ill-posed noisy learning objective and further prevent overfitting. Indeed, entropy regularization is favorable in situations for which the joint distribution, $p(x, y)$, is mis-specified (Grandvalet & Bengio, 2004) which further underpins the motivation of pseudo-labeling as an apt basis for regularization of the GCE loss.

Pseudo-labeling and data augmentation often go hand-in-hand. Data augmentation serves dual purpose as a generic regularizer to mitigate over-fitting of noisy labels (Zhang et al., 2018) as well as provides additional information about the vicinity or neighborhood of the training examples which is formalized by Vicinal Risk Minimization (Chapelle et al., 2001). These augmented examples can be seen as drawn from a vicinity distribution of the training examples to enlarge support of the training distribution such that samples in the vicinity share the same class but does not model the relation across examples of different classes (Zhang et al., 2018). Therefore, data augmentations approximate samples of nearby elements of the data manifold where the difference, $\xi(x) = \mathcal{A}(x) - x$, approximates elements of its tangent space (Athiwaratkun et al., 2019). In this way, ECR can loosely be seen as generating a set of stochastic differential constraints at each optimization step of the classification task loss. While stronger augmentation can enrich the vicinity distribution, augmentation methods such as MixUp (Zhang et al., 2018) and RandAugment (Cubuk et al., 2020) can overly degrade training examples and drift off the data manifold (Hendrycks et al., 2020). When learning with noise, it is therefore essential to leverage an augmentation process rich in variety but which also preserve the image semantics and local statistics such as AugMix (Hendrycks et al., 2020) so as to minimize the additional strain on an already ill-posed noisy learning objective. Consistent with this understanding, although RandAugment has been successfully leveraged in semi-supervised learning (Berthelot et al., 2020; Kurakin et al., 2020; Xie et al., 2019a), our experiments with RandAugment proved unsuccessful for extreme levels of label noise. Moreover, AugMix augmentation used together with the Jensen-Shannon consistency loss endows trained models with far superior model robustness to corrupted data in deployment as shown in Table 7.

## 3 Related Work

Some methods for learning with noise attempt to improve noisy learning performance head-on by leveraging augmentation as a strong regularizer to mitigate memorization of corrupted labels (Zhang et al., 2018) while others attempt to refurbish corrupted labels to control the accumulation of noise from mislabeled data (Song et al., 2019). A recent theme in learning with noisy labels has been to transform the learning with noise problem into a semi-supervised one by removing the labels of training data determined to be corrupted to form the requisite dichotomy of clean labeled data and a pool of unlabeled data (Nguyen et al., 2020; Li et al., 2020); then directly applying semi-supervised approaches such as MixMatch (Berthelot et al., 2019) and MeanTeacher (Tarvainen & Valpola, 2017). Other methods go so far as to require trusted human verified data and combine re-weighting with re-labeling into a meta optimization approach (Zhang et al., 2020).

Semi-supervised learning algorithms have advanced considerably in recent years, making heavy use of both data augmentation and consistency regularization. MixMatch (Berthelot et al., 2019) proposed a low-entropy label-guessing approach for augmented unlabeled data and mixes labeled and unlabeled data using MixUp. In MixMatch, pseudo-label targets are formed by averaging label distributions produce by the model on samples drawn from the vicinity distribution ($\frac{1}{K}\sum_K p_\theta(y|\mathcal{A}(x))$). However, this averaging requires artificial sharpening to generate low-entropy pseudo-labels. From the MAP estimation perspective, sharpening does not add auxiliary a priori knowledge for the optimization step but rather prescribes a desirable property of the model generated label distribution. Indeed, our experiments with the use of artificial label sharpening in RTE resulted in failed training at high levels of label noise and subsequent related work recognized that stronger augmentations can result in disparate predictions so their average may not generate meaningful targets (Berthelot et al., 2020). ReMixMatch (Berthelot et al., 2020) introduced augmentation anchoring which aims to minimize the entropy between label distributions produced by multiple weak and strong data augmentations of unlabeled data using a control theory augmentation approach. While pseudo-label guessing and augmentation anchoring motivate the utility of multiple augmentations of the same data, our proposed ECR for learning with noise differs in the following important ways: ECR does not use distribution alignment for "fairness", distribution averaging, or label-sharpening; ECR forms pseudo-label targets using an exponential average of model weights and is batch-synchronized with the GCE task loss term. Finally, the recent work, FixMatch (Kurakin et al., 2020), proposes a simplified semi-supervised approach where the consistency regularization term uses hard pseudo-labeling for low-entropy targets together with a filtering step to remove low-confidence unlabeled examples but does not leverage multiple strong augmentations.

## 4 EXPERIMENTS

In this section we analyze the performance of RTE against various uniform noise configurations for both symmetric and asymmetric settings. For asymmetric noise, we test both the traditional configuration (Patrini et al., 2016), typically reported by related works, and an additional configuration defined by (7) which is in the spirit of (Lee et al., 2019), where we define the asymmetric noise structure using the confusion matrix of a trained shallow network. In all of these experiments, RTE outperforms existing methods. Finally, we perform additional ablation studies to better understand the contribution and synergy of the terms in equation (6) as well as to probe the efficacy of ECR.

In our experiments we consider the standard CIFAR-10, CIFAR-100, and ImageNet datasets (Krizhevsky, 2009; Deng et al., 2009). CIFAR-10 and CIFAR-100 each contain 50,000 training and 10,000 test images of 10 and 100 classes, respectively; and ImageNet contains approximately 1,000,000 training images and 50,000 validation images of 1000 classes. Additionally, we test networks trained with noisy labels against unforeseen input corruptions using CIFAR-10-C (Hendrycks & Dietterich, 2019) which was constructed by corrupting the original CIFAR-10 test set with a total of 15 noise, blur, weather, and digital corruptions under different severity levels and intensities. Classifier performance is averaged across these corruption types and severity levels to yield *mean corruption error* (mCE). Since CIFAR-10-C is used to measure network behavior under data shift, these 15 corruptions are not included during the training procedure. Here, CIFAR-10-C helps to establish a rigorous benchmark for image classifier robustness to better understand how models trained with noisy data might perform in safety-critical applications.

To mitigate the sensitivity of experimental results to empirical, and perhaps arbitrary, choices of hyperparameters, we present additional results that leverage Population Based Training (PBT) (Jaderberg et al., 2017; Li et al., 2019) which is a simple asynchronous optimisation algorithm that jointly optimize a population of models and their hyperparameters. In particular, PBT discovers a per-epoch *schedule* of hyperparameter settings rather than a static fixed configuration used over the entirety of training. These PBT schedules, for example, allow task loss $\mathcal{L}_q$ to vary between CE and MAE loss dynamically during training and similarly the number of ECR terms $N^*$ can be modulated to realize a form of curriculum learning. Moreover, for our purposes, PBT schedules also provide a form of quasi-ablation study, as optimization of the task-loss parameter $q$, the number of ECR terms $N^*$, and the ECR weight $\lambda_{\text{ECR}}$ allows for the realization of a simplified loss which forgos these components if determined maximally beneficial. We find, as in other studies, that this joint optimization of hyperparameter schedules typically results in faster wall-clock convergence and higher final performance. (Ho et al., 2019; Li et al., 2019).

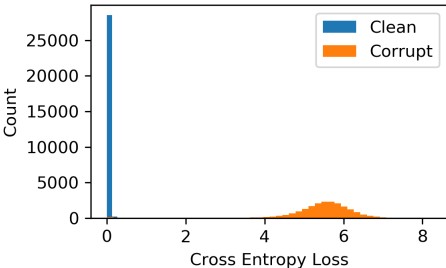 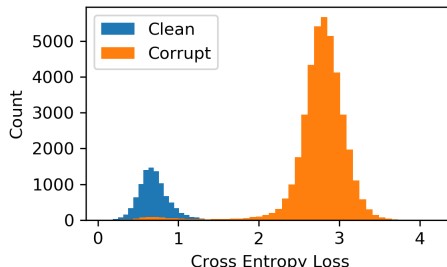

Figure 1: Loss distributions for clean labels versus corrupt labels on CIFAR-10 with 40% label noise (left) and 80% label noise (right). All losses are computed with respect to the labels used during training, which mimics a realistic setting (no access to clean labels).

## 4.1 UNIFORM SYMMETRIC NOISE

**Training Setup.** Please see Section C in the Appendix.

**Baselines**. To best interpret the effectiveness of RTE, we compare our results to many techniques for learning with noise (Table 1). A description of each baseline method can be found in Appendix B. Only two of these references provide ImageNet results trained with label noise (Table 2).

**Results**. Experimental results with uniform symmetric noise for both CIFAR-10 and CIFAR-100 are presented in Table 1 with comparisons to related work, including current state-of-the-art methods. RTE establishes new state-of-the-art performance at all noise levels and exhibits especially large performance gaps at high noise levels. At 80% noise, previous state-of-the-art was achieved by (Arazo et al., 2019) in the case of CIFAR-10 and by (Li et al., 2020) in the case of CIFAR-100. RTE improves performance over these methods by 7.0 absolute percentage points and 6.2 absolute percentage points, respectively. Of all of these works, only two report results on ImageNet training with noisy labels. These are included alongside RTE results in Table 2, where once again we see that RTE performs favorably, improving state-of-the-art performance in terms of both top-1 accuracy and top-5 accuracy. As in (Arazo et al., 2019) and (Li et al., 2020), we also include loss distributions over clean and corrupt labels in Figure 1. Here we can see that RTE prevents rote memorization of noisy labels. Moreover, Table 7 shows that RTE retains strong corruption robustness with an mCE of 12.05% and 13.50% at noise ratios of 40% and 80% respectively, as measured using CIFAR-10-C. Put in context, experiments summarized in Table 7 indicate that even with extreme levels of mislabeled training data, RTE trained models have lower corruption error than models trained using standard methods using clean data.

## 4.2 UNIFORM ASYMMETRIC NOISE

**Training Setup**. For consistency, uniform asymmetric noise experiments use the same hyperparameter configurations outlined for uniform symmetric noise. Here we test RTE performance using both the traditional asymmetric noise configuration (Patrini et al., 2016) typically reported by related works defined by Equation 8 in Section G of the Appendix as well as an additional configuration in the spirit of (Lee et al., 2019) where we define the asymmetric noise structure using the confusion matrix of a trained shallow network defined by Equation 7 in Section D of the Appendix.

The asymmetric noise defined by Patrini et al. (2016) in equation (8) does not corrupt all classes but rather attempts to capture a noise process whereby labelers confuse specific pairs of classes which by some is argued to be more realistic in practice (Han et al., 2018; Ren et al., 2018). We additionally consider a rich noise structure by training a shallow classifier (ResNet-10) on clean CIFAR-10 and use the resulting confusion matrix of this model to define the noise structure in equation (7). For example, this asymmetric noise process readily captures the phenomenon that objects on blue backgrounds are often confused (e.g. birds, ships, and airplanes) and its natural asymmetry where $p(\tilde{y}_i = \text{SHIP}|y_i = \text{AIRPLANE}) = 0.2772$ while $p(\tilde{y}_i = \text{AIRPLANE}|y_i = \text{SHIP}) = 0.4603$ (locations $[1, 9]$ and $[9, 1]$ in Eq. 7). Dataset statistics are provided for an instance of CIFAR-10 with

Table 1: Test accuracy on CIFAR-10 and CIFAR-100 under uniform symmetric label noise. Results in parentheses are upper bounds since they were computed using lower noise levels. The results for Reed-Hard, S-Model (Goldberger & Ben-Reuven, 2016), Forward T and Co-Teaching are from Nguyen et al. (2020) and the results for MixUp and Meta-Learning are from Li et al. (2020). RTE provides better robustness to label noise than all other methods. Higher is better.

| Method | # Params | CIFAR-10 Noise Ratio | | CIFAR-100 Noise Ratio | |
|---|---|---|---|---|---|
| | | 40% | 80% | 40% | 80% |
| *(Prior Work)* | | | | | |
| Reed-Hard (Reed et al., 2014) | – | 69.66 | – | 51.34 | – |
| S-Model | – | 70.64 | – | 49.10 | – |
| MentorNet PD (Jiang et al., 2018) | 84M | 77 | 33 | 56 | 14 |
| Forward T (Patrini et al., 2016) | – | 83.25 | 54.64 | 31.05 | 8.90 |
| Open Set (Wang et al., 2018) | – | 78.15 | – | – | – |
| Rand Weights (Ren et al., 2018) | 36.4M | 86.06 | – | 58.01 | – |
| Bi-Level (Jenni & Favaro, 2018) | 11.2M | 89 | – | 61.6 | – |
| GCE (Zhang & Sabuncu, 2018) | 21.8M | (87.12) | (64.07) | (61.77) | (29.16) |
| Co-Teaching (Han et al., 2018) | – | 81.85 | 29.22 | 55.95 | 23.22 |
| MixUp (Zhang et al., 2018) | – | – | (71.6) | – | (30.8) |
| SELFIE (Song et al., 2019) | – | 86.5 | – | 62.9 | – |
| RoG (Lee et al., 2019) | – | 81.83 | – | 55.68 | – |
| M-DYR-H (Arazo et al., 2019) | 11.2M | – | 86.6 | – | 48.2 |
| PENCIL (Yi & Wu, 2019) | 21.8M | – | – | 69.12 | "fail" |
| Meta-Learning (Li et al., 2019) | – | – | (77.4) | – | (42.4) |
| SELF (Nguyen et al., 2020) | 25.0M | 93.70 | 69.91 | 71.98 | 42.09 |
| DivideMix (Li et al., 2020) | 11.2M | 94.9 | 79.8 | 75.2 | 60.2 |
| *(Our Work)* | | | | | |
| RTE (Manual) | 13.1M | 94.84 | 93.09 | 76.70 | 64.02 |
| RTE (PBT) | 13.1M | **95.52** | **93.64** | **77.44** | **66.43** |

Table 2: Validation accuracy on ImageNet with 40% uniform symmetric label noise.

| | MentorNet | SELF | RTE |
|---|---|---|---|
| # Params | 59M | 25.0M | 25.6M |
| Top-1 Acc | 65.1 | 71.31 | **74.79** |
| Top-5 Acc | 85.9 | 89.92 | **91.26** |

asymmetric label noise prescribed according to equation (7) with a uniform noise ratio of 60% in Table 9 of Appendix G .

**Baselines**. In the case of asymmetric noise as defined in Patrini et al. (2016), by equation (8), we compare the performance of RTE against existing work. A brief description of each baseline method can be found in Appendix B. In the case of asymmetric noise structure as defined in equation (7), to our knowledge, prior work does not exist, and we report RTE performance at varied noise levels.

Table 3: Test accuracy on CIFAR-10 with asymmetric noise as defined in Patrini et al. (2016) by equation (8). Higher is better.

| | Noise Ratio: 40% | | | | |
| --- | --- | --- | --- | --- | --- |
| | GCE | SELF | PENCIL | DivideMix | RTE |
| Test Acc | 64.79 | 89.07 | 91.16 | 93.4 | **94.49** |

Table 4: RTE test performance on CIFAR-10 for different ratios of uniform asymmetric noise defined according to equation (7). Sharp declines in accuracy begin to occur at 65% noise due to more AUTOMOBILE images labeled as TRUCK, than actual TRUCK images labeled as TRUCK, and so on.

| | Noise Ratio | | | | | |
| --- | --- | --- | --- | --- | --- | --- |
| | 20% | 40% | 60% | 65% | 70% | 72% |
| $\uparrow$ Test Acc | 95.34 | 94.82 | 93.99 | 80.55 | 72.12 | 59.70 |
| $\downarrow$ mCE | 11.22 | 11.89 | 13.73 | 25.44 | 33.61 | 44.87 |

**Results**. The results for asymmetric noise as presented in related works defined in Patrini et al. (2016) by equation (8) with a uniform noise ratio of 40% are shown in Table 3 along side the performance of related methods. Again, RTE improves the state-of-the-art performance in this category, with a 1.1 absolute percentage point increase over (Li et al., 2020).

Test accuracy for different level of asymmetric noise using $C$ defined by (7) are shown in Table 4. Even with 60% noise ratio, RTE achieves 93.99% test accuracy. The first significant decline in accuracy occurs around a 65% asymmetric noise ratio, when the majority labels in a class are corrupted labels from another class. That is, for $F_{p=0.65}$ with $C$ defined by (7), there are more AUTOMOBILE images labeled as TRUCKs, than actual TRUCK images labeled as TRUCK.

### 4.3 ABLATION STUDIES

We perform various ablation studies to better understand the contribution of each term in equation (6), probe the efficacy of ECR, and compare with alternative regularization approaches. Our ablation results are presented in Table 5. These ablation studies use the training configurations defined in section 4.1 unless otherwise stated. We perform a component analysis where we remove one component at a time from equation 6 to better understand the performance contributions of each term. Removal of any term degrades performance. We also test alternative consistency regularization approaches using label guessing as proposed in MixMatch (Berthelot et al., 2019) and augmentation anchoring from ReMixMatch (Berthelot et al., 2020) which both underperform by significant margins compared to ECR. Moreover, our results show significant benefits in the use of EMA whereas performance degrades with the augmentation anchoring approach consistent with prior work (Berthelot et al., 2019). Additionally, we test if label sharpening could benefit ECR, but we find that the artificial sharpening process amplifies noisy pseudo-labels early in training and learning collapses for high noise ratios. Similarly, we find the strong linear chains of augmentations performed by RandAugment lead to training instabilities. Figure 2 summarizes the comparison of ECR to a traditional semi-supervised approach where a larger batch size is used for unsupervised regularization terms. This comparison indicates improved noisy learning performance with batch synchronization and repeated augmentation over larger batch sizes with single augmentations, validating the use of ECR for learning with noise.

### 5 CONCLUSION

We introduced robust temporal ensembling (RTE), which unifies semi-supervised consistency regularization and noise robust task loss as an effective approach for learning with noisy labels. Rather than discarding noisy labels and applying semi-supervised methods, we successfully demonstrated a new approach for learning with noise which leverages all the data together without the need to filter,

Table 5: Ablation study. Test accuracy reported from CIFAR-10 with 80% noisy labels. Label guessing (Berthelot et al., 2019) and augmentation anchoring (Berthelot et al., 2020) use a sharpening temperature of $T = 0.5$ as recommended in the associated related works.

| Ablation | Test Acc | Ablation | Test Acc |
|---|---|---|---|
| RTE | 93.09 | Label Guessing, $K = 2$ | 79.09 |
| No ECR ($\lambda_{\mathrm{ECR}} = 0$) | 61.91 | Aug. Anchoring, $K = 2$ | 83.59 |
| No GCE ($q = 0$) | 76.08 | Aug. Anchoring, $K = 4$ | 83.24 |
| No JSD ($\lambda_{\mathrm{JSD}} = 0$) | 90.37 | Aug. Anchoring, $K = 6$ | 83.20 |
| with ECR, $N^* = 2$, no EMA | 67.23 | Aug. Anchoring, $K = 2$, EMA | 77.38 |
| with ECR, $N^* = 2$, no batch-sync | 88.46 | ECR with Label Sharpening | fail |
| with ECR, $N^* = 2$, batch-sync | 91.90 | ECR with RandAugment | fail |

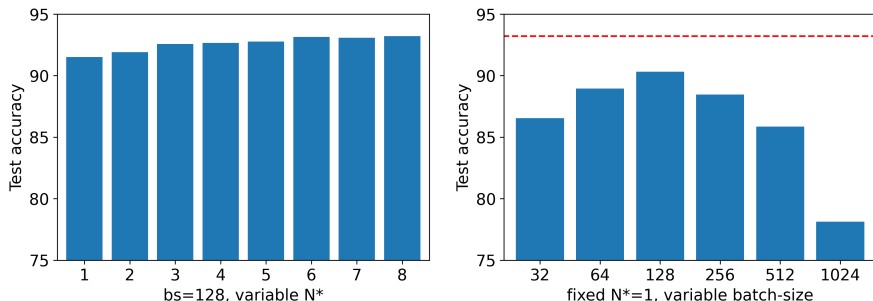

Figure 2: RTE ablation study using CIFAR-10 with uniform symmetric noise ratio of 80%. Left: the ECR batch entries are shared with the task loss and the batch size is fixed at 128, while the number of ECR terms ($N^*$) is varied. Right: 1 ECR term is used with varying ECR batch size, using batch entries that are distinct from the task loss (analogous to a more traditional semi-supervised approach). The dashed red line on the right is the ECR baseline established using $N^* = 8$.

refurbish, or abstain from noisy training examples. Through various experiments, we showed that RTE performs quite well in practice, advancing state-of-the-art performance across the CIFAR-10, CIFAR-100, and ImageNet datasets by 7.0, 6.2, and 3.5 absolute percentage points, respectively. Moreover, experiments summarized in Tables 4 and 7 show that despite significant label noise, RTE trained models retain lower corruption error on unforeseen data shifts than models trained using standard methods on clean data. Finally, the results of numerous ablations summarized in section 4.3 validate the composition of loss terms and their combined efficacy over alternative methods. In future work, we are interested in the application of RTE for different modalities such as natural language processing and speech where label noise can be more pervasive and subjective.

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

## A  APPENDIX: HYPERPARAMETERS

Table 6: Manual hyperparameter configurations for uniform symmetric noise experiments. The two $\lambda$ values are $\lambda_{\mathrm{JSD}}$ and $\lambda_{\mathrm{ECR}}$.

|  | Wt. Decay | $\beta$ | Dropout | BS | LR | $\lambda$ | $q$ | $N^*$ | $\alpha$ |
|---|---|---|---|---|---|---|---|---|---|
| CIFAR-10 | .001 | .9 | .01 | 128 | $\cos\left(\frac{7\pi k}{16K}\right)$ | 12, 1 | $\sin\left(\frac{13\pi k}{16K}\right)$ | 10 | .99 |
| CIFAR-100 | .0005 | .9 | .01 | 128 | 0.04 | 5, 3 | 0.3 | 8 | .99 |
| ImageNet | .001 | .9 | .00 | 256 | .1, .01, .001 | 12, 10 | 0.3 | 3 | .99 |

## B  APPENDIX: BASELINES FOR TABLE 1

In this section we provide a brief summary of the baseline methods in the main text:

**Reed et al. (2014)** introduce two methods for achieving prediction consistency, one based on reconstruction and one based on bootstrapping, and demonstrated empirically that bootstrapping leads to better robustness to label noise. **Goldberger & Ben-Reuven (2016)** model the correct label as latent and having gone through a parameterized corruption process. Expectation maximization is used to estimate both the parameters of the corruption process and the underlying latent label. **Jiang et al. (2018)** introduce the idea of *learning* a curriculum-learning strategy with a *mentor* model to train a *student* model to be robust to label noise. **Patrini et al. (2016)** estimate the noise transition matrix (under the assumption of feature independent noise) and show that, given the *true* noise transition matrix, optimizing for the true underlying labels is possible. **Wang et al. (2018)** introduce an iterative scheme that combines 1. outlier detection in feature space (acting as a proxy to noisy-label detection), 2. a Siamese network (taking either a clean, clean pair or a clean, noisy pair) to encourage separation, and 3. sample reweighting based on clean vs. noisy confidence levels in order to effectively filter out noisy labels during training. They focus primarily on *open-set* noise, but they also report performance of their system when used in the *closed-set* setting. **Ren et al. (2018)** use a meta-learning approach to dynamically weight examples to minimize loss *using a set of validation examples with clean labels*, however they also report a competitive baseline using a randomized weighting scheme which requires no clean validation set. **Jenni & Favaro (2018)** formulate example weighting as a bilevel-optimization problem, in which performance on a validation set is maximized with respect to example weights, subject to the constraint that the model maximizes performance on the training set; and they argue that this approach should lead to better generalization when label noise is present. **Zhang & Sabuncu (2018)** introduce a loss function that is a generalization of cross-entropy loss and mean absolute error, which is beneficial since each exhibits distinct desirable properties: cross-entropy exhibits better gradient properties for learning, while mean absolute error exhibits better theoretically-grounded robustness to noisy labels. **Han et al. (2018)** leverage co-teaching such that two networks are trained together, in which each network 1. identifies high-confidence examples, 2. passes this information in a message to its peer, and 3. leverages the incoming message to optimize using the examples selected by its peer. **Zhang et al. (2018)** train using convex combinations of both input images and their labels, arguing that this approach makes it more difficult for the network to memorize corrupt labels. **Song et al. (2019)** measure label consistency throughout training in order to determine which samples are 'refurbishable', and these samples are then 'corrected' by replacing their ground-truth label with the most frequently-predicted label. **Lee et al. (2019)** do not modify the training process of the underlying neural network but instead form a generative model over the final (pre-softmax) features of the neural network, and this generative distribution along with Bayes rule is then used to estimate a more robust conditional distribution over the label. **Arazo et al. (2019)** fit a beta mixture model over the *loss* using two mixture components, representing *clean* and *noisy* labels, and each sample's underlying component probabilities are used to weight each sample's contribution during training. They combine this approach with MixUp (Zhang et al., 2018). **Yi & Wu (2019)** maintain a direct estimate of a distribution over true underlying labels during training, and train the parameters of a neural network by minimizing reverse KL divergence (from the model's predicted distribution to these true-label estimates). Mean-

while a 'compatibility loss' is introduced to ensure that the estimated label distribution stays close to the noisy labels provided with the training set. **Li et al. (2019)** subject a student model to artificial label noise during training and take alternating gradient steps and maintain a teacher model that is not subjected to such noise. Here, alternating gradient steps are taken to 1. minimize classification loss and 2. minimize the KL divergence from the student's predicted distributions to the teacher's predicted distributions. **Nguyen et al. (2020)** use discrepancy between an ensemble-based teacher model and labels to identify and filter out incorrect labels, and continue to leverage these samples without the labels. This is done in a semi-supervised fashion by maintaining consistency between the teacher's predictions and the student's predictions. **Li et al. (2020)** maintain two networks and for each network models *loss* using a mixture of Gaussians with two components (*clean* and *noisy*). Each network estimates which samples belong to each component, and the *other* network then uses the *clean* samples in a supervised manner along with the *noisy* labels in a semi-supervised manner.

## C    APPENDIX: UNIFORM SYMMETRIC NOISE EXPERIMENTAL SETUP

For CIFAR-10, we leverage equation (2) with $C_{i \neq j}^{10} = \frac{1}{9}$ and we employ a 28-layer residual network (He et al., 2016) with a widening factor of 6 (WNR 28x6) (Zagoruyko & Komodakis, 2016), a dropout rate of 0.01 (Srivastava et al., 2014), $\alpha = 0.99$, AugMix with a mixture width and severity of 3, a batch size of 128, and 300 epochs of training. We optimize using SGD with Nesterov momentum of 0.9 (Sutskever et al., 2013), a weight decay of 0.001, and a cosine learning rate (Loshchilov & Hutter, 2017) of $0.03 \cdot \cos(7\pi k/16K)$, where $k$ is the current training step and $K$ is the total number of training steps. The RTE loss function (6) is configured with static $\lambda_{\text{JSD}}$, $\lambda_{\text{ECR}}$ and $N^*$ of 12, 1, and 10, respectively, whereas $q$ is scheduled according to $0.6 \cdot \sin(13\pi k/16K)$ (which assigns small $q$-values in early training epochs, reaches a maximum of $q = 0.6$ after 180 epochs, and decreases to $q = 0.33$ over the remaining 120 epochs). For CIFAR-100, the setup is similar, but different hyperparameters are used; details are included in the Appendix in Table 6. In addition to manual configurations, we consider PBT with a population size of 35 to optimize learning rate, weight decay, $q$, $\lambda_{\text{JSD}}$, $\lambda_{\text{ECR}}$ and $N^*$. Fastidious readers will find the complete PBT configuration defined in Appendix F. For ImageNet, ResNet50 is used and trained with SGD for 300 epochs with a stepped learning rate of 0.1, 0.01 and 0.001 which begin at epochs 0, 100 and 200 respectively. ImageNet hyperparameters are also included in the Appendix in Table 6.

## D    APPENDIX: CONFUSION MATRIX FOR UNIFORM ASYMMETRIC NOISE

$$C = \begin{pmatrix} .0000 & .0396 & .2475 & .0594 & .0594 & .0396 & .0495 & .0693 & .2772 & .1584 \\ .1765 & .0000 & .0294 & .0000 & .0000 & .0000 & .0294 & .0000 & .1765 & .5882 \\ .1745 & .0000 & .0000 & .1544 & .1879 & .1074 & .2617 & .0872 & .0268 & .0000 \\ .0388 & .0116 & .1473 & .0000 & .1240 & .3682 & .1899 & .0853 & .0155 & .0194 \\ .0303 & .0000 & .2197 & .1667 & .0000 & .0606 & .2879 & .2121 & .0227 & .0000 \\ .0324 & .0000 & .1435 & .4676 & .1019 & .0000 & .1204 & .1157 & .0093 & .0093 \\ .0536 & .0179 & .3571 & .3036 & .1071 & .0714 & .0000 & .0536 & .0179 & .0179 \\ .0704 & .0000 & .0986 & .1268 & .3803 & .1831 & .0986 & .0000 & .0000 & .0423 \\ .4603 & .0952 & .0794 & .0476 & .0317 & .0000 & .0476 & .0317 & .0000 & .2063 \\ .1711 & .5132 & .0263 & .0526 & .0263 & .0132 & .0658 & .0395 & .0921 & .0000 \end{pmatrix} \quad (7)$$

## E    APPENDIX: PERFORMANCE DATA ON CIFAR-10-C

Table 7: RTE mean corruption error on CIFAR-10-C for models trained at various uniform symmetric noise ratios. Baseline reference values for 'Standard' and 'AugMix' mCE are reported from Hendrycks et al. (2020) using WRN 40x2 on clean data. Lower is better.

|  |  |  | Noise Ratio | | |
| --- | --- | --- | --- | --- | --- |
|  | Standard | AugMix | 0% | 40% | 80% |
| ↓ mCE | 26.9 | 11.2 | 11.5 | 12.05 | 13.50 |

## F APPENDIX: PBT EXPERIMENTS

Table 8: PBT sampling configuration for CIFAR-10 and CIFAR-100. We used a population size of 35, and permutation interval of 2 epochs. In the case a member inherits another checkpoint, each hyperparameter is resampled from its distribution with $p = 0.25$ or is multiplied with $w \sim \text{Uniform}(0.8, 1.2)$ within its parameter range with $p = 0.75$. In the case of $N^*$, the previous/next hyperparameter from the ordered list is selected.

| Parameter | Sample distribution |
|---|---|
| Batch size | 128 |
| Dropout | 0.01 |
| $\beta$ | 0.9 |
| $\alpha$ | 0.99 |
| LR | Uniform(0.00001, 0.1) |
| weight decay | Uniform(0.00005, 0.002) |
| $q$ | Uniform(0.0, 1.0) |
| $\lambda_{\text{JSD}}$ | Uniform(0.0, 20.0) |
| $\lambda_{\text{ECR}}$ | Uniform(0.0, 5.0) |
| $N^*$ | Uniform$\{3, ..., 10\}$ |

## G APPENDIX: UNIFORM ASYMMETRIC NOISE ON CIFAR-10

The matrix $C$ defines the noise structure for uniform asymmetric noise on CIFAR-10 with following labels: AIRPLANE, AUTOMOBILE, BIRD, CAT, DEER, DOG, FROG, HORSE, SHIP, TRUCK.

$$C = \begin{pmatrix} 0 & 0 & 0 & 0 & 0 & 0 & 0 & 0 & 0 & 0 \\ 0 & 0 & 0 & 0 & 0 & 0 & 0 & 0 & 0 & 0 \\ 1 & 0 & 0 & 0 & 0 & 0 & 0 & 0 & 0 & 0 \\ 0 & 0 & 0 & 0 & 0 & 1 & 0 & 0 & 0 & 0 \\ 0 & 0 & 0 & 0 & 0 & 0 & 0 & 1 & 0 & 0 \\ 0 & 0 & 0 & 1 & 0 & 0 & 0 & 0 & 0 & 0 \\ 0 & 0 & 0 & 0 & 0 & 0 & 0 & 0 & 0 & 0 \\ 0 & 0 & 0 & 0 & 0 & 0 & 0 & 0 & 0 & 0 \\ 0 & 0 & 0 & 0 & 0 & 0 & 0 & 0 & 0 & 0 \\ 0 & 1 & 0 & 0 & 0 & 0 & 0 & 0 & 0 & 0 \end{pmatrix} \tag{8}$$

## H APPENDIX: EXTENDED DATA AND ANALYSIS

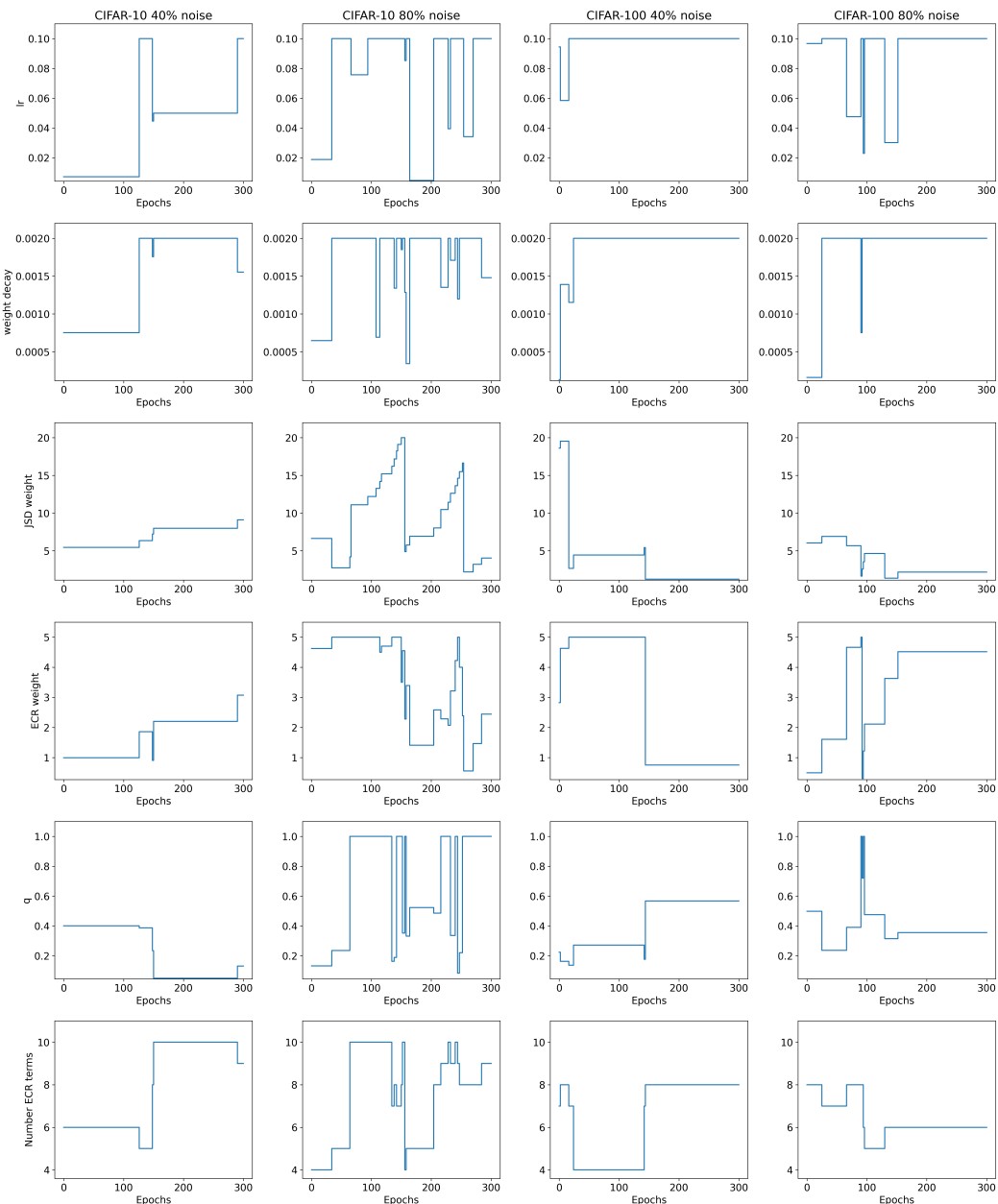

Figure 3: Parameter schedules for $lr$, weight decay, JSD weight $\lambda_{JSD}$, ECR weight $\lambda_{ECR}$, $q$ and $N^*$ for CIFAR-10 and CIFAR-100 with 40% and 80% uniform symmetric noise rates.

Table 9: Overview of class distribution of total and correct labels after sampling noisy CIFAR-10 training labels with asymmetric noise defined by equation (7) with a uniform 60% noise ratio.

|  | # samples | % samples | # correct labels | % correct labels |
|---|---|---|---|---|
| AIRPLANE | 5578 | 11% | 1958 | 35% |
| AUTOMOBILE | 4069 | 8% | 2003 | 49% |
| BIRD | 6023 | 12% | 2017 | 33% |
| CAT | 6205 | 12% | 2038 | 33% |
| DEER | 5056 | 10% | 1986 | 39% |
| DOG | 4480 | 9% | 1977 | 44% |
| FROG | 5476 | 11% | 2019 | 37% |
| HORSE | 4130 | 8% | 2028 | 49% |
| SHIP | 3896 | 8% | 2024 | 52% |
| TRUCK | 5087 | 10% | 1950 | 38% |
| TOTAL | 50000 | 100% | 20000 | 40% |

Table 10: RTE test accuracy and mean corruption error (mCE) on CIFAR-10 and CIFAR-10-C, respectively. In this experiment, fixed batch size of $bs = 128$ is used and the number of ECR terms, $N^*$ is varied. Training configuration of these data is described in section 4.1. Test accuracy is presented in Figure 2 (left).

| CIFAR-10 | | | | | | | |
|---|---|---|---|---|---|---|---|
| Fixed batch-size: 128 | | | | | | | |
| Uniform Symmetric Noise: 80% | | | | | | | |
| Vary the number of ECR terms: $N^*$ | | | | | | | |
| 1 | 2 | 3 | 4 | 5 | 6 | 7 | 8 |
| ↑ Test Acc  91.51 | 91.90 | 92.57 | 92.65 | 92.77 | 93.14 | 93.09 | 93.21 |
| ↓ mCE  15.32 | 14.87 | 13.74 | 13.90 | 13.84 | 13.48 | 13.67 | 13.66 |

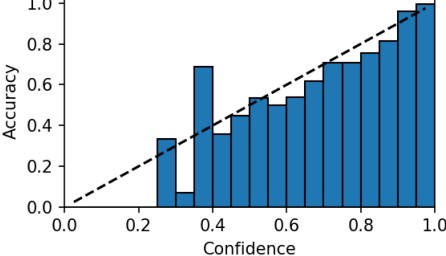 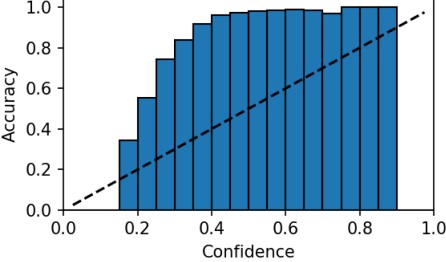

Figure 4: Reliability diagrams for RTE training models on CIFAR-10 with 40% uniform label noise (left) and 80% label noise (right). Perfectly calibrated models follow the black line, whereas over-confident models lie below and under-confident models lie above. This figure indicates our RTE trained model is well calibrated when trained with 40% label noise, while (perhaps justifiably) conservative when trained with a more extreme level of 80% label noise.

Table 11: RTE test accuracy and mean corruption error (mCE) on CIFAR-10 and CIFAR-10-C, respectively. In this experiment a single consistency loss term is used and vary the batch size of that term. This experiment with varying batch size is analogous to a more traditional semi-supervised approach where large batch size is used for unsupervised loss terms. Training configuration for these data is described in section 4.1. Test accuracy is presented in Figure 2 (right).

| | CIFAR-10 | | | | | |
|---|---|---|---|---|---|---|
| | Fixed ECR terms: $N^* = 1$ | | | | | |
| | Uniform Symmetric Noise: 80% | | | | | |
| | Vary the batch size: | | | | | |
| | 32 | 64 | 128 | 256 | 512 | 1024 |
| ↑ Test Acc | 86.54 | 88.95 | 90.32 | 88.46 | 85.87 | 78.13 |
| ↓ mCE | 19.77 | 17.78 | 16.41 | 18.20 | 20.42 | 28.57 |

