# OpenReview forum: "Robust Temporal Ensembling"
_ICLR.cc/2021/Conference — Reject_

### Official Review · AnonReviewer2 · 2020-10-13
**Review: Robust Temporal Learning**

**Rating:** 6
**Confidence:** 4

**Review:**

##########################################################################
Summary:

Real-world data contains noise in the annotated labels. To mitigate, the authors propose a supervised learning approach, Robust Temporal Ensembling (RTE). RTE combines 1) task loss correction, which is a generalized cross entropy loss, 2) different augmentations resulting from AugMix technique and the Jensen-Shannon divergence (JSD), 3) the ensemble consistency regularization and pseudo labeling.

##########################################################################

Reasons for score:

Overall, I vote for accepting. The idea of improving the robustness predictions in noisy labeled data is interesting. My major concerns are on the motivation and the clarity of the idea at some places. Hopefully the authors will address these concerns in the rebuttal period.
##########################################################################


Pros:
+ Overall, the paper is well written with minimal to no grammatical errors, easy to follow and understandable.

+ The approach is well motivated, clearly describes the use of loss functions and the ensemble consistency, etc. Positions the proposed approach on how the existing techniques are combined and further improved.

+ The paper is strong in terms of empirical evidence when combined with multiple existing techniques.

+ Ensuring the true effective noise ratio is mathematically interesting, as opposed to the state-of-the-art practices.

+ The idea of forgoing the identification and filtering/fixing noisy labels is an encouraging piece of contribution which further strengths this line of research.

+ Solid experimental results, although having results on other benchmarks does not hurt. However the ImageNet results dominate.

Cons:
- Motivation for this work can be further improved (In the Introduction in Paragraph 1). I see the citation for Fe-Fei Li work on Imagenet.

“Jia Deng, Wei Dong, Richard Socher, Li-Jia Li, Kai Li, and Li Fei-Fei. Imagenet: A large-scale hi- erarchical image database. In 2009 IEEE conference on computer vision and pattern recognition, pp. 248–255, 2009.”

- Maybe this can be used early on and you can highlight the problem of label noise in the following way, if not exactly, “Amazon Mechanical Turk spent 49k people spread across 167 countries over 2.5 of years. Yet, Imagent has label noise ….”
The long history of noise robust learning is missing the approaches of abstention, falling under identifying and filtering or fixing the incorrect labels.

“Sunil Thulasidasan, Tanmoy Bhattacharya, Jeff A. Bilmes, Gopinath Chennupati, Jamal Mohd-Yusof: Combating Label Noise in Deep Learning using Abstention. ICML 2019: 6234-6243”

“Lihong Li, Michael L. Littman, and Thomas J. Walsh. 2008. Knows what it knows: a framework for self-aware learning. In Proceedings of the 25th international conference on Machine learning (ICML '08).”

- Novelty of the method is suboptimal. The proposed technique is a combination of methods in literature.
Although true effective noise ratio is interesting, it does not really make much difference in ImageNet kind of large number (1000) of classes (1/1000 * 0.8) .

- “In semi-supervised learning techniques it is typical to leverage a larger ...” the above sentence does not flow well, maybe worth re-written.

- Too many hyper-parameters

- “... while q is prescribed an ad-hoc schedule ...” does not make sense, modify please

- Is this effective noise ratio kept exactly 40/80% for the baselines unlike the common practice? From the results, it does not appear that way, i) please clarify, ii) if not setting those noise ratios might benefit the paper even more because the baselines might perform even worse.

- The temporal ensembling part is little vague in the current form of the paper, probably hiding/missing from section 3.2.3. Can you be more specific with the details for this?

- Also, is it possible to stud the certainty of predictions, in terms of calibration error or some other metric? If they can be squeezed in the paper somehow, that will further strengthen the paper. Moreover, one can actually understand the effect of ensembles on the robustness of predictions in noisy environments.

---

> ### Author Response · Authors · 2020-11-19
> **Response**
>
> Thank you for your review, and for your many positive comments regarding clarity, strong experimental results, and strengthening this line of research by forgoing label filtering/fixing.
>
> Here we focus on the criticisms and questions. **Bold** is used to highlight the most important points.
>
> > Motivation for this work can be further improved (In the Introduction in Paragraph 1). I see the citation for Fe-Fei Li work on Imagenet.
>
> Thank you for pointing this out – motivation was clear in our mind, but we of course agree that it should also be completely obvious to any reader. We have added your suggested motivation to the introduction, and we have added other motivations as well.
>
> > The long history of noise robust learning is missing the approaches of abstention, falling under identifying and filtering or fixing the incorrect labels.
>
> Thank you – we have added this in the introduction.
>
> > Novelty of the method is suboptimal. The proposed technique is a combination of methods in literature.
>
> - **Please note that ECR differs from alternative consistency regularization strategies in a number of other ways.**
> - **We have updated the manuscript both to describe differences between ECR and prior methods and to include additional ablation studies that further emphasize this point (Table 5).**
> - Additionally, please note that alternative SSL consistency losses do not perform as well as ECR for learning with noisy labels.
>
> > Is this effective noise ratio kept exactly 40/80% for the baselines unlike the common practice? From the results, it does not appear that way, i) please clarify, ii) if not setting those noise ratios might benefit the paper even more because the baselines might perform even worse.
>
> No, they are not, and you are correct – some baselines would perform "even worse". This is highlighted in the caption of Table 1, which indicates that "results in parentheses are upper bounds since they were computed using lower noise levels".
>
> > The temporal ensembling part is little vague in the current form of the paper, probably hiding/missing from section 3.2.3. Can you be more specific with the details for this?
>
> Thank you for pointing this out; we added a sentence in Section 2.2.2 to clarify this point. To summarize, a copy of parameters is made upon initialization, and from that point onward, this copy is updated using an exponential moving average of the model parameters as training iterations progress.
>
> > [Typos / incorrect grammar]
>
> Thank you – fixed
>
> > Also, is it possible to stud the certainty of predictions, in terms of calibration error or some other metric? If they can be squeezed in the paper somehow, that will further strengthen the paper. Moreover, one can actually understand the effect of ensembles on the robustness of predictions in noisy environments.
>
> Thank you for this suggestion. We have added loss distributions for clean vs. noisy labels to the main text, and we have added reliability diagrams (to gauge calibration error) to the appendix.

---

> > ### Comment · AnonReviewer2 · 2020-11-24
> > **Thanks for the update and fixes**
> >
> > Thank the authors for the clarifications. However, still believe it is an incremental paper, therefore will stick with my current scores.

---

### Official Review · AnonReviewer3 · 2020-10-18
**Official Blind Review #3**

**Rating:** 5
**Confidence:** 5

**Review:**

Summary: this paper proposes a method for learning with label noise. The proposed method combines three techniques: GCE from robust loss literature, AugMix from data augmentation literature, and Mean Teacher from semi-supervised learning literature. This paper shows that the combination of these methods are effective for label noise learning.

Strength: this paper studies an important problem; the combination of existing methods is intuitive, and each method plays an important role; the paper is mostly well-written and easy to follow.

Weakness:
1. The paper claims to "introduce a new ensemble-based form of consistency regularization which leverages multiple augmentations of the same images". However, this strategy has already been used by methods such as MixMatch (MixMatch: A Holistic Approach to Semi-Supervised Learning).
2. The proposed method seems to be a rather ad-hoc combination of several existing methods (GCE, AugMix, Mean Teacher), hence the technical novelty is limited. It is fine as long as the experimental results are strong, which I am not fully convinced.
3. The comparison with some previous methods on CIFAR seem to be unfair. For example, DivideMix uses a 18-layer PreAct ResNet, whereas this paper uses a 28- layer Wide ResNet. It is important to make sure that all previous methods are compared under a fair setting.
4. How important is AugMix to the model's performance? What if a different augmentation is used?
5. The results in Table 5 (robustness to data shift under label noise) is expected because AugMix does not consider label noise. Hence it is hard to justify the value of this experiment.
6. The propose method is not validated on real-world noisy datasets, such as the widely used WebVision, Food-101, or Clothing1M. Experiments on synthetic noisy datasets alone cannot fully justify the effectiveness of the method, because real-world noise can be more complicated.

---

> ### Author Response · Authors · 2020-11-19
> **Response**
>
> Thank you for your review and for your positive comments regarding the importance of the problem and the clarity of the paper.
>
> Here we focus on the criticisms. **Bold** is used to highlight the most important points.
>
> > The paper claims to "introduce a new ensemble-based form of consistency regularization which leverages multiple augmentations of the same images". However, this strategy has already been used by methods such as MixMatch (MixMatch: A Holistic Approach to Semi-Supervised Learning).
>
> **Although many consistency losses share common elements, ECR is significantly different. We added substantial clarification to highlight this point (in the Methods and Related Work sections).**
>
> **To highlight this further, we also included several additional ablation studies. These can be found in Table 5, and include approaches based on MixMatch and its follow up, ReMixMatch.** The best result using these approaches is 83.59%, in comparison to 93.09% achieved by RTE.
>
> > The comparison with some previous methods on CIFAR seem to be unfair. For example, DivideMix uses a 18-layer PreAct ResNet, whereas this paper uses a 28- layer Wide ResNet. It is important to make sure that all previous methods are compared under a fair setting.
>
> **Please note that among the 17 prior methods in Table 1, architectures include**
> - **PreAct ResNet-18, 11.2M params (e.g., DivideMix).**
> - **WRN 28x10, 36.4M params (e.g., Rand. Weights)**
> - **ResNet-34 with shake-shake regularization, 27.0M params (e.g., SELF)**
> - **ResNet-101, 44.5M params (e.g., MentorNet)**
> - **DenseNet, unspecified param count (e.g., RoG)**
> - **and others.**
>
> **Meanwhile, RTE uses WRN 28x6, with 13.1M params.**
>
> **Given the diversity in prior work, we feel that it is unfair to suggest that we should have chosen the DivideMix architecture for our experiments.**
>
> To clarify this in the manuscript, we added parameters counts to Table 1 and Table 2.
>
> Finally, we would like to ask if you believe it is possible that this small difference in capacity (11.2M params vs. 13.1M params) is responsible for the stark difference between DivideMix and RTE at 80% label noise on CIFAR-10 (79.8% accuracy vs. 93.64% accuracy)?
>
> > How important is AugMix to the model's performance? What if a different augmentation is used?
>
> We have experimented with AugMix and RandAug, and we have added the latter to the ablation study (Table 5). We found that RandAug leads to unstable training.
>
> > The results in Table 5 (robustness to data shift under label noise) is expected because AugMix does not consider label noise. Hence it is hard to justify the value of this experiment.
>
> Our best interpretation of this statement is *it's not worth demonstrating that your method maintains robustness to variations in the input distribution.*
>
> If this is accurate, we respectfully disagree. If this is inaccurate, can you please elaborate?
>
> > The propose method is not validated on real-world noisy datasets, such as the widely used WebVision, Food-101, or Clothing1M. Experiments on synthetic noisy datasets alone cannot fully justify the effectiveness of the method, because real-world noise can be more complicated.
>
> **Including more experiments is always better, but we simply do not have enough time for these experiments, and we have already surpassed the experiments in most prior work:**
> - **We include ImageNet experiments, which was done by only 2 of the 17 prior works in Table 1 (as noted in the manuscript)**
> - **We further included asymmetric noise results using a confusion matrix built from real-world datasets using a shallow neural network**

---

> > ### Comment · AnonReviewer3 · 2020-11-25
> > **Thanks for the response, but my concerns still remain.**
> >
> > I appreciate the author's response. However, some of my major concerns still remain. In particular, it is not clear to me how much performance improvement comes from increased model size and better augmentation. Furthermore, there lacks experiments on real-world noisy datasets. Therefore, I'll keep my original score.

---

### Official Review · AnonReviewer4 · 2020-10-26
**The main contribution of this paper is  to combine the generalized cross entropy loss from robust classification with temporal consistency losses from semi-supervised learning for robust classification to noisy labels. It is a purely empirical paper. It's however useful and offers promising gains, for the adopted noise model.**

**Rating:** 5
**Confidence:** 4

**Review:**

Overall impression
This submission deals with robust supervised learning in the presence of noisy labels. The label noise is modeled using a probabilistic (and conditionally independent) transition matrix that changes the label of one class to another one. In order to classify with noise, the network is trained with a mixture of three known losses including: 1) generalized cross entropy (GCE) rejects the outlier labels, 2) JSD divergence to assure the soft-max distribution matches the augmented data distributions, and 3) an ensemble consistency regularization (ECR) that penalizes the inconsistencies of the augmented data based on the mean teachers. Experiments with CIFAR-10, CIFAR-100, and ImageNet classification indicate substantial gains compared with state-of-the-art alternatives.

Strong points:
- The empirical study of combining three different metrics is extensive and useful; the gains are also significant; ablation study is also useful

Weak points:
- The contribution is rather incremental, combining three known loss metrics in outlier detection and semi-supervised learning
- The motivation behind eq.5 is not very clear. Why is it needed to use augmented data in eq. 5? why not simply using mean-teachers (ensemble of previous network weights) to penalize the consistency between the predicted labels and the noisy ones, in the same spirit as mean teachers?


Suggestions:
- The paper would improve if Table 1 and the ablation study can include the results for ImageNet as a more realistic dataset as well .
- What is index i running over in eq (5)?
- A lot of details about experiments are provided in the main paper. For more clarify, those could be moved to appendix, and more intuition and explanation about the reason for adopting the three loss components would be more useful. perhaps toy examples with diagrams could be helpful.

---

> ### Author Response · Authors · 2020-11-19
> **Response**
>
> Thank you for your review and for your positive comments regarding our experiments and ablation study.
>
> Here we focus on the criticisms. **Bold** is used to highlight the most important points.
>
> > The contribution is rather incremental, combining three known loss metrics in outlier detection and semi-supervised learning
>
> - **We have updated the manuscript both to describe differences between ECR and prior methods and to include additional ablation studies that further emphasize these differences.**
> - **Please note that alternative SSL consistency losses do not perform as well as ECR for learning with noisy labels.**
>
> > The motivation behind eq.5 is not very clear. Why is it needed to use augmented data in eq. 5? why not simply using mean-teachers (ensemble of previous network weights) to penalize the consistency between the predicted labels and the noisy ones, in the same spirit as mean teachers?
>
> This is a good question and we have updated our manuscript to be sure to address it.
>
> **First, please note that augmentations can be viewed as a regularizer to prevent overfitting, which is important in the case of noisy data - to highlight this, we have added additional motivation in Section 2.3.**
>
> **In addition, please note that mean teacher uses augmentation.** They showed that augmentation significantly improves performance ([1], Table 5 on page 11).
>
> **Finally, please note that ECR differs from alternative consistency regularization strategies in a number of other ways. We have highlighted these differences in the updated manuscript as well.** As an example, consider SELF, which uses mean teacher but achieves 69.9% vs. RTE's 93.09% for CIFAR10 at an 80% noise level.
>
> [1] A. Tarvainen and H. Valpola, Mean teachers are better role models, NeurIPS, 2017.
>
> > The paper would improve if Table 1 and the ablation study can include the results for ImageNet as a more realistic dataset as well.
>
> We understand this sentiment, but please understand that this is not practical because of resource constraints. Further, we would like to highlight that we have already gone beyond most prior work. **Of the 17 prior works listed in Table 1, fewer than half include any ablation study at all, and only 2 of the 17 report ImageNet results at all.**
>
> > What is index i running over in eq (5)?
>
> To be precise, we could write A_i(x), where A is a sampled augmentation function. We omitted the subscript for brevity, under the understanding that augmentations are inherently random.
>
> > A lot of details about experiments are provided in the main paper. For more clarify, those could be moved to appendix, and more intuition and explanation about the reason for adopting the three loss components would be more useful. perhaps toy examples with diagrams could be helpful.
>
> Thank you for this suggestion. We have moved experimental results to the appendix to make room, and we have elaborated substantially on each loss term, their motivations, and the differences between ECR and other consistency-regularization methods.

---

### Official Review · AnonReviewer1 · 2020-10-28
**Robust Temporal Ensembling**

**Rating:** 6
**Confidence:** 3

**Review:**

**Summary:**
This work aims to train models with noisy training labels.
1. This work introduces Robust Temporal Ensembling, which is composed of ideas introduced in prior work:
 - Noise-robust task loss (Generalized cross entropy from (Zhang & Sabuncu, 2018))
 - Augmentation (JSD loss from AugMix (Hendrycks* et al., 2020))  Minimizes the KL divergence between each of the three model outputs (original image + two augmentations) and the average of those three outputs.
 - Ensemble consistency regularization (ECR) combines (Tarvainen & Valpola, 2017) which uses a moving average of model parameters and the consistency regularization formalized in (Laine & Alia, 2017) to compare the difference in outputs of the teacher and a set of augmentations from AugMix.
2. These three factors contribute to the final loss.  The remainder of the work demonstrates the advantage of this combined loss through a barrage of experiments.
 - They show that on CIFAR-10, CIFAR-100 (with 40% and 80% corruption) and ImageNet (with 40%) corruption, their approach outperforms all baselines, often by a considerable margin.
 - They also explore the impact of different types of noise (class-symmetric, asymmetric, and varying degrees of noise)
 - Ablation experiments show that the main novelty (ECR) plays the largest role in the performance of the model, but GCE is also needed to get state of the art performance on CIFAR-10 (80%)
 - Finally, they show that using multiple augmentations in ECR in a single batch can considerably improve performance.

**Positives:**
 - This approach clearly outperforms all the compared baselines (to my knowledge, they aren’t missing any comparisons)
 - The experiments are quite extensive.
 - The exposition clearly explains how this work relates to prior work and how it composes those ideas into the final model.
**Negatives:**
 - This work leans heavily on prior work.  See summary above for how it relates to prior work.
**Recommendation:**
The main contribution of this work is the ECR term, which uses an idea from semi-supervised learning to match the prediction to a “mean teacher”.  By extending it to use multiple augmentations, they improve results over using a single augmentation (without mean teacher Laine & Alia, 2017) or without augmentations (vs. a mean teacher, Tarvainen & Valpola, 2017).
The question is whether this combination of loss terms is a significant enough contribution.  I think it’s borderline, but since the performance gain is so significant and since I can’t think of any way to improve the work, I’ll lean toward accepting.




**Minor comments:**
Eq (5) is missing a  )

---

> ### Author Response · Authors · 2020-11-19
> **Response**
>
> Thank you for your review, and for your many positive comments regarding clarity and the thoroughness of our experiments.
>
> Here we focus on the sole criticism, which is that it leans heavily on prior work. **Bold** is used to highlight the most important points.
>
> - **We have updated the manuscript both to describe differences between ECR and prior methods and to include additional ablation studies that further emphasize these differences.**
> - Please note that ECR is the most important component of RTE and that alternative SSL consistency losses do not perform as well for learning from noisy labels.
> - As an example, consider SELF, which uses mean teacher but is considerably outperformed by ECR.
>
> We also fixed Equation 5. Thank you.

---

### Author Response · Authors · 2020-11-19
**General Rebuttal Overview and Manuscript Changes**

We thank all of the reviewers for their thoughtful feedback.

A clear message from the reviewers was that our simplified presentation was transparent but led to a need for improved focus on motivations and contributions. We have updated the draft submission accordingly, and we provide a short summary here:

- **We ask whether we can leverage the underlying mechanisms of semi-supervised learning such as entropy regularization *without* following the trend in recent work: discarding labels.**

- **We show that it is indeed possible to leverage all of the data together, without the need to filter noisy training examples, and, through a series of ablation experiments, we show that our form of consistency regularization is key. This is shown in Table 5, where we can see that ECR significantly outperforms other variants such as those from MixMatch and ReMixMatch.**

A general list of changes to the manuscript is as follows:

- Introduction: added prior work on abstention-based methods, additional motivation and clarity around central contribution
- Methods: expanded descriptions of loss components; better emphasis on differences between ECR and previous consistency regularization approaches; added insights for choices of loss terms and the synergy of their combination for learning with noise
- Related Work: Moved to after Methods for better context; refactored to address reviewer feedback regarding motivation
- Experiments: As suggested by reviewers, some experimental details were moved to the appendix; added parameter counts to Table 1 and Table 2; updated ImageNet results as those numbers were erroneously taken from a clean experiment; addition of  Figure 1: loss distributions of clean labels and corrupt labels based on feedback from R2; added additional ablation studies to strengthen motivation and better communicate ECR effectiveness over alternative methods and configurations
- Appendix: As requested by reviewers, extended model performance analysis has been added.  Figure 4, reliability diagrams displaying RTE trained model calibration

---

### Decision · Program_Chairs · 2021-01-07
**Final Decision**

**Decision:**

Reject

**Comment:**

Reading the paper and the reviews themselves, I found myself conflicted about this work:

- Multiple reviewers commented that this is a rather incremental piece of work, given that it's a rather straightforward combination of existing losses/models.
- On the other hand, there is admittedly value in (1) realizing that this combination is meaningful (2) understanding the meaningful ways in which these work or do not work with ablation studies.
- I am not quite satisfied that the datasets and experiments in this work represent in any meaningful way real world noise. However, it does appear that the authors ran experiments on common benchmarks using common protocols so there's only so much that they themselves can be blamed for.
- Tangentially, I am somewhat surprised about the relatively good ImageNet performance of this method. I suspect the combination of this being done with uniform noise rather than structured noise is helping quite a bit.

All in all, this work is certainly interesting enough, but the results are just not quite compelling enough to pass the bar.